# A Method for Locational Risk Estimation of Vehicle–Children Accidents Considering Children’s Travel Purposes

**DOI:** 10.3390/ijerph192114123

**Published:** 2022-10-29

**Authors:** Kojiro Matsuo, Kosuke Miyazaki, Nao Sugiki

**Affiliations:** 1Department of Architecture and Civil Engineering, Toyohashi University of Technology, Toyohashi 441-8580, Japan; 2Department of Civil Engineering, National Institute of Technology, Kagawa College, Kagawa 761-8058, Japan

**Keywords:** elementary school students, locational accident risk, travel purpose, Empirical Bayes estimation

## Abstract

The reduction in locational traffic accident risks through appropriate traffic safety management is important to support, maintain, and improve children’s safe and independent mobility. This study proposes and verifies a method to evaluate the risk of elementary school students-vehicle accidents (ESSVAs) at individual intersections on residential roads in Toyohashi city, Japan, considering the difference in travel purposes (i.e., school commuting purpose; SCP or non-school commuting purpose: NSCP), based on a statistical regression model and Empirical Bayes (EB) estimation. The results showed that the ESSVA risk of children’s travel in SCP is lower than that in NSCP, and not only ESSVAs in SCP but also most ESSVAs in NSCP occurred on or near the designated school routes. Therefore, it would make sense to implement traffic safety management and measures focusing on school routes. It was also found that the locational ESSVA risk structure is different depending on whether the purpose of the children’s travels is SCP or NSCP in the statistical model. Finally, it was suggested that evaluation of locational ESSVA risks based on the EB estimation is useful for efficiently extracting locations where traffic safety measures should be implemented compared to that only based on the number of accidents in the past.

## 1. Introduction

Road traffic accidents are one of the major causes of death and disability among children over the world [1]. The threat of being involved in a traffic accident contains not only the risk of direct physical damage but also the long-term impact on children’s lifestyles. Namely, the greater the concern parents/guardians feel about traffic accidents involving their children, the less likely they are to allow their children to play freely outside or to go out independently without adults [2,3,4,5,6]. In fact, it has been reported that children’s opportunities to walk and cycle, and play outside, as well as children’s independent mobility (CIM), have decreased in the 21st century in some countries [7,8,9,10,11,12]. The decrease in daily walking and cycling opportunities leads to children’s lack of physical activity [7]. Children’s opportunities for physical activity are important for their physical maturation and the form of normal bone and behavior, and physical inactivity is widely recognized as one of the major risk factors in human health [8,13]. It has been argued that CIM is also important in fostering children’s autonomy and in curbing car-independent transportation style which reduces time lost by parents taking their children as well as traffic congestion [10]. Therefore, the reduction in traffic accident risks and threats through appropriate traffic safety management is very important in urban planning to support, maintain, and improve children’s safe and independent mobility (CSIM).

In traffic safety management, it is important to properly assess the risk of traffic accidents at each area or location, especially for identifying areas/locations where countermeasures should be implemented and verifying the effectiveness of those countermeasures. In previous studies on the evaluation of accident risk by areas/locations, statistical model analyses (so-called safety performance functions: SPF) have often been used to clarify the effects of various explanatory variables on the number of accidents [14]. Table 1 lists the main previous studies using a statistical modeling approach for the assessment of accident risk involving children by areas/locations [15,16,17,18,19,20]. The aggregation units for accident risk assessment are diverse, ranging from the microscopic level such as intersections and road segments to the macroscopic level such as prefectures. While these previous studies have made important contributions to the methodology for assessments of accident risks involving children by areas/locations, there have been issues to be dealt with.

The first is the consideration of the children’s travel purposes. The risk of children walking and cycling being involved in accidents may be different between traveling for school commuting purposes (SCP) and traveling for non-school commuting purposes (NSCP). In Japan, children are expected to be safer on the road in SCP than in NSCP, because many elementary schools have adopted a group walking system, called “Shudan-togeko”, to and from schools, and parents/guardians and district communities provide generous safety support [21,22]. In recent years, several countries, including the United Kingdom, also have begun to implement group walking initiatives, called “Walking bus”, to school [21]. Thus, accident risk assessments that take into account the differences between children’s travel in SCP and in NSCP are internationally important. Inada et al. [20] analyzed traffic accidents involving elementary school students and junior high school students by prefecture in Japan, dividing the accidents into those involving the students in SCP and NSCP, and found that the number of killed or seriously injured (KSI) cases during travels in SCP was approximately 30–40% lower than that during travels in NSCP. However, the study was conducted only at the macro level (i.e., prefecture-level), and, to the best of the author’s knowledge, there have been no studies in which whether the children’s purpose was SCP or NCSP was taken into account in a micro level (such as intersections or road segments level) analysis.

As the second issue, the accident risk of each location may be affected by a great variety of factors specific to the location. It is difficult to incorporate all of the factors as explanatory variables in a statistical model, and the effects of factors other than the explanatory variables are represented as the error term with a probability distribution. Therefore, in assessing accident risks at different locations, it is necessary to properly consider the balance between the average effects of explanatory variables and the effect of location-specific characteristics. While the Empirical Bayesian (EB) estimation method has been proposed and used in the context of a before–after study on the evaluation of traffic safety measures [23,24,25,26,27,28,29,30], it is also applicable to the purpose of extracting locations where countermeasures should be implemented by properly evaluating the accident risks at individual locations [31,32,33,34,35,36,37]. Although there have been some previous studies applying the EB method to evaluate the locational risks of accidents involving general pedestrian [30,33], no studies applying the EB method to assess the locational risks of accidents involving children have been seen.

Therefore, this study proposes and verifies a method to evaluate the risk of vehicle–children accidents at individual intersections on residential roads in Toyohashi city, Japan, considering the difference in travel purposes (i.e., SCP or NSCP), based on a statistical regression model and EB estimation.

## 2. Materials and Methods

### 2.1. Target Area

The target area was the entire Toyohashi City, Aichi Prefecture, Japan. There are 52 elementary schools in the city (see Figure 1), and the number of elementary school students is approximately 20,000 as of 2018. Almost all the students walk to school, except for about 10 students who cycle to school.

### 2.2. Data

#### 2.2.1. Traffic Accident Data

The accident data used in this study were police-reported elementary school student-vehicle accidents (ESSVAs: injury accidents in which the first or second party was an elementary school student while walking or cycling) that occurred in Toyohashi City over a 12-year period from 2009 to 2020 (totally 638 ESSVAs). The individual accident data include the children’s travel purpose (i.e., SCP or NSCP). Figure 1 shows the distribution of the locations of the ESSVAs by the children’s travel purpose.

The ESSVAs which occurred within a radius of 50 m from the center of each community street intersection were extracted as “intersection ESSVAs”, which were 360 ESSVAs (56.4% of all the ESSVAs). The community street intersections were defined as intersections with three or more legs of community streets other than “national expressways”, “general national highways”, “major regional roads”, “general prefectural roads”, and “major general roads” in the road network data of Zmap-Area II (ZENRIN corp.). Of the 7719 target intersections, one ESSVA occurred at 296 (3.8%) intersections, two ESSVAs at 24 (0.31%) intersections, three ESSVAs at four (0.052%) intersections, and four ESSVAs at one (0.013%) intersection, during the 12-year period.

We used the 299 ESSVAs in the first nine years as the “risk estimation period” to estimate the ESSVA risk at each intersection, and the 61 ESSVAs in the last three years as the “evaluation period” to evaluate the results of the estimation.

#### 2.2.2. Risk Exposure Data

Accident risk exposure is an important factor for the statistical model analysis of locational accident risks, and some studies have made efforts to measure exposure for evaluating the locational risk of accidents involving pedestrians [38,39,40,41]. In this study, the exposure should be the number of passings or activities of elementary school students at each intersection. However, it is difficult to measure these data over the wide city area. As shown in Table 1, the previous studies have tried to estimate exposure by children population [18,20], by questionnaire surveys asking about travel routes [16,17], by an observation survey of travel mode to schools [19], and by a journey allocation model based on shortest routes [16].

On the other hand, the author and the Toyohashi City Board of Education have collaboratively georeferenced the designated school route data (see Figure 1) and the gathering point data, with the number of elementary school students for each school commuting group in the city. In this study, based on the designated school route data (2018 version), we calculated the following indices as surrogate variables for ESSVA risk exposure at the intersections [42]:Number of elementary school students on the surrounding school routes (NSSR15, 50, 100): the number of elementary school students on the designated school route that passes within a radius of 15 m, 50 m, or 100 m from the center of each target intersection;Number of elementary school students at the surrounding gathering points (NSGP15, 50, 100): the number of elementary school students at the gathering points located within a radius of 15 m, 50 m, or 100 m from the center of each target intersection;Distance to the nearest school route (DSR): the distance from the center of each intersection to the nearest designated school route.

Note that there are no major differences in the designated school route data in recent years.

#### 2.2.3. Other Data

The ESSVA risk at an intersection is also may be greatly affected by the vehicle traffic volume [16,43,44]. Therefore, in order to take into account the effect of vehicle traffic volume at each target intersection, this study used probe vehicle data corrected by Pioneer Corp. for the years 2016–2019. The validity of the probe vehicle data has been confirmed in our previous studies [30]. The probe vehicle traffic volumes of all legs in the direction of inflow to the intersection were summed up to obtain the probe vehicle traffic volume for each target intersection, which was used as a variable to represent the vehicle traffic volume.

As other explanatory variables for road traffic environment conditions, we used the number of legs for each intersection, the presence or absence of traffic signals, the urban area land use type by 1/10 mesh (100 m mesh) of the National Land Information, whether each intersection is within the densely inhabited district (DID), and the distance to the nearest park.

### 2.3. Statistical Model

In this study, the locational accident risk is defined as the expected value of the number of ESSVAs by intersections in a specific period. Therefore, a count data model is used as the statistical model. The Poisson regression model is the simplest count data model. However, because the error structure (i.e., Poisson distribution) has a strong limitation that the variance equals the expected value, the negative binomial (NB) regression model, which is one of the extensions of the Poisson regression model, is commonly applied in recent accident count data modeling [14]. Additionally, in this study, the NB regression model is used.

To the best of the author’s knowledge, the EB estimation has been found in the field of traffic accident risk assessment since Gipps [23], Abbess et al. [24], and Jarrett et al. [31] in the early 1980s, and later it was well-organized by Hauer [25] in the context of before/after study of traffic safety measures. In order to summarize the method for assessing accident risk considering location-specific factors using the EB estimation, it is first explained that the Poisson regression model can be extended to the NB regression model by assuming the property that the expected value in the Poisson regression model itself varies stochastically according to the gamma distribution. Then, the EB method is presented for the evaluation of the risk of accidents at each location, taking into account the location-specific effects that cannot be explained by the average effects (fixed effects) of the explanatory variables considered in the model.

#### 2.3.1. Poisson Regression Model

Suppose that the number of accidents $Y_{i}$ at location i during a certain period of time follows a Poisson distribution with an expected value $\mu_{i}$ that is a function of the explanatory variable vector $x_{i}$ and the corresponding parameter vector b:(1)$$Y_{i}\sim \mathrm{Poisson}\left(\mu_{i}\right)=\mathrm{Poisson}\left(x_{i},\;b\right).$$

Thus, the probability that $Y_{i}$ accidents occur at location i during the period is
(2)$$P_{\mathrm{Poi}}\left(Y_{i}=y_{i}|\;\mu_{i}\right)=\frac{\mu_{i}^{y_{i}}}{y_{i}!}\exp\left(-\mu_{i}\right),$$
and the expected value and variance of $Y_{i}$ are
(3)$$E\left(Y_{i}|\;\mu_{i}\right)=V\left(Y_{i}|\;\mu_{i}\right)=\mu_{i},$$
where
(4)$$\mu_{i}=\exp\left(x_{i}b\right)=\exp\left(b_{0}+\underset{k}{\sum }b_{k}x_{ik}\right).$$

#### 2.3.2. Accident Risk Evaluation That Considers Location-Specific Factors

Assuming that the expected value of the Poisson regression model described above is not completely determined only by the function of explanatory variables and parameters, but is also affected by factors specific to location i, and then assuming as an alternative to the expected value $\mu_{i}$ the random variable $\lambda_{i}$ which follows a gamma distribution with the shape parameter α=ϕ and the rate parameter $\beta =\phi \mu_{i}^{-1}$:(5)$$\lambda_{i}\sim \mathrm{Gamma}\left(\alpha =\phi ,\beta =\phi \mu_{i}^{-1}\right).$$

The probability that $\lambda_{i}$ be a certain value $r_{i}$ is
(6)$$P_{\mathrm{Gam}}\left(\lambda_{i}=r_{i}|\;\alpha =\phi ,\;\beta =\phi \mu_{i}^{-1}\right)=\frac{\phi^{\phi }\mu_{i}^{-\phi }}{\Gamma \left(\phi \right)}r_{i}^{\phi -1}\cdot \exp\left(-\phi \mu_{i}^{-1}r_{i}\right),$$
and the expected value and variance of $\lambda_{i}$ are
(7)$$E\left(\lambda_{i}\right)=\frac{\phi }{\phi \mu_{i}^{-1}}=\mu_{i}=\exp\left(x_{i}b\right),$$
(8)$$V\left(\lambda_{i}\right)=\frac{\phi }{\phi^{2}\mu_{i}^{-2}}=\frac{\mu_{i}^{2}}{\phi }.$$

The structure that the expected value of the number of accidents $\lambda_{i}$ is determined following the gamma distribution, and the realization $y_{i}$ of the number of accidents $Y_{i}$ occurs following the Poisson distribution with $\lambda_{i}$ as the parameter can be interpreted as a hierarchical model. Therefore, under the information that $y_{i}$ accidents actually occurred at location i, the probability that $\lambda_{i}$ is a certain value $r_{i}$ can be obtained by Bayes’ theorem:(9)$$P\left(\lambda_{i}=r_{i}|\;Y_{i}=y_{i};\;\mu_{i},\phi \right)=\frac{P_{\mathrm{Poi}}\left(Y_{i}=y_{i}|\;\mu_{i}\right)P_{\mathrm{Gam}}\left(\lambda_{i}=r_{i}|\;\phi ,\phi \mu_{i}^{-1}\right)}{P\left(Y_{i}=y_{i}|\;\mu_{i},\phi \right)}.$$

Here, $\mu_{i}$ and ϕ are the prior parameters. Furthermore, the gamma distribution is the natural conjugate distribution of the Poisson distribution, thus it is known that the posterior distribution is also gamma, so that:(10)$$P\left(\lambda_{i}=r_{i}|\;Y_{i}=y_{i};\;\mu_{i},\;\phi \right)=P_{\mathrm{Gam}}\left(\lambda_{i}=r_{i}|\;\phi +y_{i},\phi \lambda \mu_{i}^{-1}+1\right),$$
and its expected value is
(11)$$E\left(\lambda_{i}|\phi +y_{i},\phi \mu_{i}^{-1}+1\right)=\frac{\phi +y_{i}}{\phi \mu_{i}^{-1}+1}.$$

The expected value of $\lambda_{i}$ can be used as an accident risk evaluation index that takes into account location-specific factors.

#### 2.3.3. EB Estimation through an NB Regression Model

The accident risk evaluation index in Equation (11) depends on the prior parameters $\mu_{i}$ and ϕ. Therefore, in this study, these prior parameters are estimated by the EB method. That is, we use $\mu_{i}$ and ϕ which maximize the marginal likelihood:(12)$$P\left(Y_{i}=y_{i}|\;\mu_{i},\phi \right)=\int_{-\infty }^{\infty }P_{\mathrm{Poi}}\left(y_{i}|\;r_{i}\right)P_{\mathrm{Gam}}\left(r_{i}|\;\phi ,\phi \mu_{i}^{-1}\right)dr_{i}.$$ Expanding Equation (12) further and setting $p_{i}=\frac{\phi }{\phi +\mu_{i}}$, we obtain:(13)$$P\left(Y_{i}=y_{i}|\;\mu_{i},\;\phi \right)=\left(\begin{array}{c} y_{i}+\phi +1\\ \phi -1 \end{array}\right){\left(p_{i}\right)}^{\phi }{\left(1-p_{i}\right)}^{y_{i}},$$
indicating that $Y_{i}$ follows an NB distribution with parameters $\mu_{i}$ and ϕ:(14)$$Y_{i}\sim \mathrm{NB}\left(\mu_{i},\;\phi \right)=\mathrm{NB}\left(x_{i},\;b,\;\phi \right).$$ The expected value and variance of $Y_{i}$ are
(15)$$E\left(Y_{i}|\mu_{i},\phi \right)=\frac{1-p_{i}}{p_{i}}\phi =\mu_{i}=\exp\left(x_{i}b\right).$$
(16)$$V\left(Y_{i}|\mu_{i},\phi \right)=\frac{1-p_{i}}{p_{i}^{2}}\phi =\mu_{i}+\frac{\mu_{i}^{2}}{\phi }.$$ Here, ϕ is called the dispersion parameter. The larger ϕ, the smaller the variance (residual deviation) of $Y_{i}$. When ϕ=∞, the variance is equal to the expected value, which means that the NB distribution is consistent with the Poisson distribution.

From the above, the EB estimation of $\lambda_{i}$ is possible by estimating the NB regression model using the actual number of accidents $y_{i}$ by locations and the explanatory variable vector $x_{i}$, and then by using the variance parameter $\phi^{NB}$ and the expected number of accidents by location $\mu_{i}^{NB}$ obtained from the estimated NB regression model as prior parameters. Denoting the EB estimator of $\lambda_{i}$ as $\lambda_{i}^{EB}$, it can be shown that,
(17)$$\lambda_{i}^{EB}=\frac{\phi^{NB}+y_{i}}{\phi^{NB}{\left(\mu_{i}^{NB}\right)}^{-1}+1}=\left(\frac{1}{1+\frac{\mu_{i}^{NB}}{\phi^{NB}}}\right)\mu_{i}^{NB}+\left(1+\frac{1}{1+\frac{\mu_{i}^{NB}}{\phi^{NB}}}\right)y_{i}.$$ Therefore, it can be found that this EB estimator is a weighted average of the two variables: (1) the expected number of accidents $\mu_{i}^{NB}$, which is calculated based on the average effects of the explanatory variables in the estimated NB regression model; and (2) the number of accidents $y_{i}$ that actually occurred. It can be also found that the larger the dispersion parameter $\phi^{NB}$ of the NB regression model is (i.e., the smaller the residual deviation by the NB regression model is), the greater the weight is given to $\mu_{i}^{NB}$. According to Ghosh and Meeden [45], “one of the main features of EB analysis is to borrow strength from the ensemble; that is, use information from similar sources in constructing estimators and predictors in addition to the most directly available source of information.”

Therefore, this study first estimated the parameter $b_{k}$ and the dispersion parameter $\phi^{NB}$ of the NB regression model shown in Equations (4) and (14), using the actual number of ESSVAs at each intersection i (explained in Section 2.2.1) as the objective variable $y_{i}$ and the risk exposure indices (explained in Section 2.2.2) and other road traffic environment conditions (explained in Section 2.2.3) as explanatory variables $x_{ik}$. Next, using the explanatory variables $x_{ik}$ and the estimated model parameter $b_{k}$, the NB estimate of the expected number of ESSVAs at each intersection, $\mu_{i}^{NB}$, was calculated by Equation (4). Then, using the NB estimate of the expected number of ESSVAs, the actual number of ESSVAs $y_{i}$, and the parameter $\phi^{NB}$, the EB estimate of the expected number of ESSVAs at each intersection, $\lambda_{i}^{EB}$, was calculated by Equation (17). This EB estimate of the expected number of ESSVAs was the risk evaluation value for each intersection.

### 2.4. Evaluation of Efficiency to Extract Risky Locations

To evaluate the usefulness of the EB estimation from the view of efficient extraction of high ESSVA risk locations, we calculated efficiency criteria by the following procedure. The concept of efficiency here is that “the higher the number of ESSVAs in the evaluation period per one extracted ESSVA risk intersection, the higher the efficiency”.

Each location was ranked in order of riskiness based on the three criteria, respectively:
*Actual number criterion*, i.e., the higher the actual number of ESSVAs $y_{i}$ in the risk estimation period (the first nine years), the higher the risk is;*NB estimate criterion*, i.e., the higher the expected number of ESSVAs $\mu_{i}^{NB}$ calculated by the NB regression model, the higher the risk is;*EB estimate criterion*, i.e., the higher the expected number of ESSVAs $\lambda_{i}^{EB}$ corrected by the EB estimation method, the higher the risk is.
Based on the rankings of the three criteria, the cumulative sums of the actual number of ESSVAs in the evaluation period (the last three years) up to the rank were, respectively calculated.The cumulative sums of the actual number of ESSVAs in the evaluation period for the three criteria were, respectively divided by the number of intersections up to the rank, which is the efficiency in this study.

Figure 2 illustrates the flow of this study described so far.

## 3. Results

### 3.1. Aggregation Analyses

Figure 3 shows the distribution of the occurrence time of ESSVAs by the children’s travel purposes, indicating that many of the ESSVAs occurred in the time period of 4:00–5:00 p.m. in NSCP. The high number of ESSVAs occurring in NSCP at this time means that elementary school students are mostly involved in an accident when being out to play or go somewhere (e.g., friends’ houses or shops) after returning home on weekdays or on holidays.

Figure 4 shows the distribution of the occurrence of ESSVAs regarding the distance from the nearest school route by the children’s travel purposes. As expected, most (roughly 80%) of the ESSVAs in SCP occurred on the school routes since children basically walk to school and walk back home along the designated school routes. On the other hand, it is a surprise that most ESSVAs in NSCP also occurred on or close to (roughly within 100 m of) the school routes. This could be an important fact and will be discussed, along with the statistical model estimation results below, in detail in Section 4.

### 3.2. Estimation Results of the NB Regression Model

Table 2 shows the estimation results of two types of final NB regression models using the numbers of ESSVAs in SCP and the numbers of ESSVAs in NSCP, at the target intersections in the risk estimation period as the objective variable. They indicated that the statistically significant explanatory variables were different between the SCP model and the NSCP model.

The parameter of the constant term in the SCP model was strongly lesser than those in the other models, indicating that children’s risk to be involved in an accident during school commuting is very lower than during other travels.

Regarding the surrogate variables for the ESSVA risk exposure, the more the number of elementary school students on the designated school route passing within the intersections (NSSR15), the higher the ESSVA risk was in the SCP model. In the NSCP model, on the other hand, the nearer the distance from the intersections to the nearest designated school route, the higher the ESSVA risk was. These results suggest that the designated school route data could be useful as surrogate variables for ESSVA risk exposure indicators.

As for the other explanatory variables, the main findings were as follows:ESSVA risk increased with vehicle traffic volume in both models, but the impact was less in the SCP model than in the NSCP model;ESSVA risk was higher at four-leg intersections than at three-leg intersections in the NCSP model, but there was no such effect of road structure in the SCP model;The SCP model showed a lower ESSVA risk at intersections near parks, but the NCSP model did not show such an effect;ESSVA risk was higher in areas with building use in both models, but was particularly strong in the SCP model;Intersections within the DID had lower ESSVA risk in the SCP model, but the effect is not shown in the NSCP model.

### 3.3. Results of the EB Estimation

Figure 5 shows the scatter charts plotting the ESSVA risks calculated by the NB regression model ($\mu_{i}^{NB}$) on the horizontal axes and those corrected by the EB estimation ($\lambda_{i}^{EB}$) on the vertical axes, according to the number of ESSVAs actually occurred during the risk estimation period shown at the top of the charts (0–3). The 45-degree line is also plotted. It can be seen that in the chart for the intersections where the actual numbers of ESSVAs were zero (the leftmost chart in Figure 5), all points were below the 45-degree line. This means that the expected numbers of ESSVAs were adjusted downward through the EB estimation because the actual numbers of ESSVAs had been smaller than the expected numbers estimated by the NB model. On the other hand, in the chart for the intersections where the actual numbers of ESSVAs were two or three (the right two charts in Figure 5), all points were above the 45-degree line. This means that the expected numbers of ESSVAs were adjusted upward through the EB estimation because the actual numbers of ESSVAs had been greater than the expected numbers estimated by the NB model. Additionally, in the chart for the intersections where the actual numbers of ESSVAs were one (the second left chart in Figure 5), the downward and upward adjustments described above were mixed. It can also be seen that even for the intersections with the same actual numbers of ESSVAs, the further to the right, the greater the difference between each point and the 45-degree line tended to be. This means that the higher the ESSVA risks calculated by the NB regression model were, the larger the magnitude of the adjustment by the EB estimation was.

Figure 6 shows the efficiencies calculated by the procedure explained in Section 2.4 based on the three criteria. It can be seen that the actual number criterion had the highest efficiency up to the 6th ESSVA risk rank, whereas the EB estimate criterion had the highest efficiency from the 7th up to about the 60th risk rank. This means that, in this study case, while using the actual number of ESSVAs was more efficient in extracting the top six risky locations, using the EB estimation was more efficient in extracting the risky locations beyond these six locations.

## 4. Discussion and Conclusions

The results of the aggregation analyses and the statistical model analysis showed that ESSVAs in SCP are less frequent than those in NSCP. This tendency is similar throughout Japan [20], and this may be due in large part to the effect of the cultural safety systems such as group school commuting (called “shudan-togeko”) and the safety supports such as crossing guard by district communities when traveling in SCP, as suggested by Waygood et al. [21] and Matsuo et al. [22] in macroscopic levels. Rothman et al. [19] showed a different result that intersections with school crossing guards had higher collision risk. However, there is a possibility that the locations where school crossing guards are located reflect the areas where many children walk, namely the risk exposure is higher resulting in higher accident frequencies. The result in this study is considered reasonable because the amount of risk exposure in SCP used in this is valid as described below.

The results of the aggregation analysis revealed that not only ESSVAs in SCP but also most ESSVAs in NSCP occurred on or near the designated school routes. The statistical model analysis showed that the number of elementary students commuting to school on each designated school route passing within a radius of 15 m from the center of each target intersection (NSSR15) was significant in the SCP model. This can be interpreted as the number of children passing through the designated school routes functioning as a risk exposure variable for ESSVAs in SCP. This is precisely because the number of children commuting to school on the school route itself is appropriate as exposure for ESSVA risk in SCP. Lee et al. [17] showed similar results using the number of children on the route to school based on a questionnaire survey as an exposure variable. On the other hand, in the NSCP model, the distance from each intersection to the nearest school route (DSR) was significant, indicating that the structure regarding ESSVA risk exposure differs depending on the children’s travel purposes. This result suggests that elementary school students tend to use the designated school routes even for NSCP because they are familiar with the school routes due to daily use in SCP, resulting that the number of ESSVAs on or near school routes would be higher. Therefore, even though the fact described above is that the majority of the ESSVAs occurred while walking and cycling for NSCP, it would make sense to implement traffic safety management and measures focusing on school routes as the Japanese national and regional government currently makes efforts [46].

As Bennet and Yiannakoulias [16] showed, ESSVA risk increased with vehicle traffic volume. However, the impact was less in the SCP model than in the NSCP model. This may be because there is a cultural system that contributes to the safety of children traveling to and from school in Japan, as mentioned above, and also because crossing guards are present especially in locations with heavy vehicle traffic. On the other hand, it suggests that traffic calming is more important for the safety of children traveling in NCSP. Many previous studies showed the safety effect of traffic calming devices on residential roads [47,48].

Although some previous studies used the number of legs of intersections but could not find a significant effect [17,18], this study found that ESSVA risk was higher at four-leg intersections than at three-leg intersections for NCSP travels. While the result that ESSVA risk was higher in areas with building use in both models is similar to the previous studies [15,16,17,19], this study additionally found that the effect was particularly strong for SCP travels. Inada et al. [20] showed that locations of DID have the effect of an increase in children’s accident risk through the prefectural level macroscopic analyses; however, this study revealed that the intersections within the DID have lower ESSVA risk for SCP travels. This may be because the size of group commuting and safety support by district communities, as described above, are more extensive in densely populated areas. As mentioned so far, the locational ESSVA risk structure, including the exposure, varies depending on whether the purpose of the children’s travels is SCP or NSCP, which is considered an important finding of this study.

Regarding the EB estimation results, it was shown that in this study case, using the EB estimate criterion was more efficient when extracting more than six risky locations. This suggested that the evaluation of locational accident risks based on the EB estimation is useful also for efficiently extracting locations where traffic safety measures for ESSVAs should be implemented compared to that only based on the number of accidents in the past, as argued in the previous studies applying EB method for other types of accidents [33,34,35,36,37]. In practice, it would be better to use both the number of accidents and the EB estimation method to identify risky locations.

Finally, this study has some limitations and further issues. The first is that the exposure data for locational ESSVA risks in NSCP were insufficient. While we used designated school route data as a surrogate variable for exposure in this study, the spatial distribution of children’s travels and activities in NSCP should be measured and used in some way (e.g., questionnaire/diary survey [15,17], observation survey [19], GPS logger survey [49], etc.).

The second is to be able to assess in detail the safety effects of group commuting and safety support by district communities. For this purpose, it is necessary to make comparisons between districts with different approaches to safety support, and between elementary school districts with and without group commuting.

Thirdly, although the same statistical modeling and predicting approach (i.e., NB regression model and EB estimation method) can be used for other regions and countries, the road traffic environment conditions and data availability differ among regions or countries. Therefore, it is necessary to consider an objective variable and a variety of explanatory variables including risk exposure indicators by tailoring them to those conditions.

## Figures and Tables

**Figure 1 ijerph-19-14123-f001:**
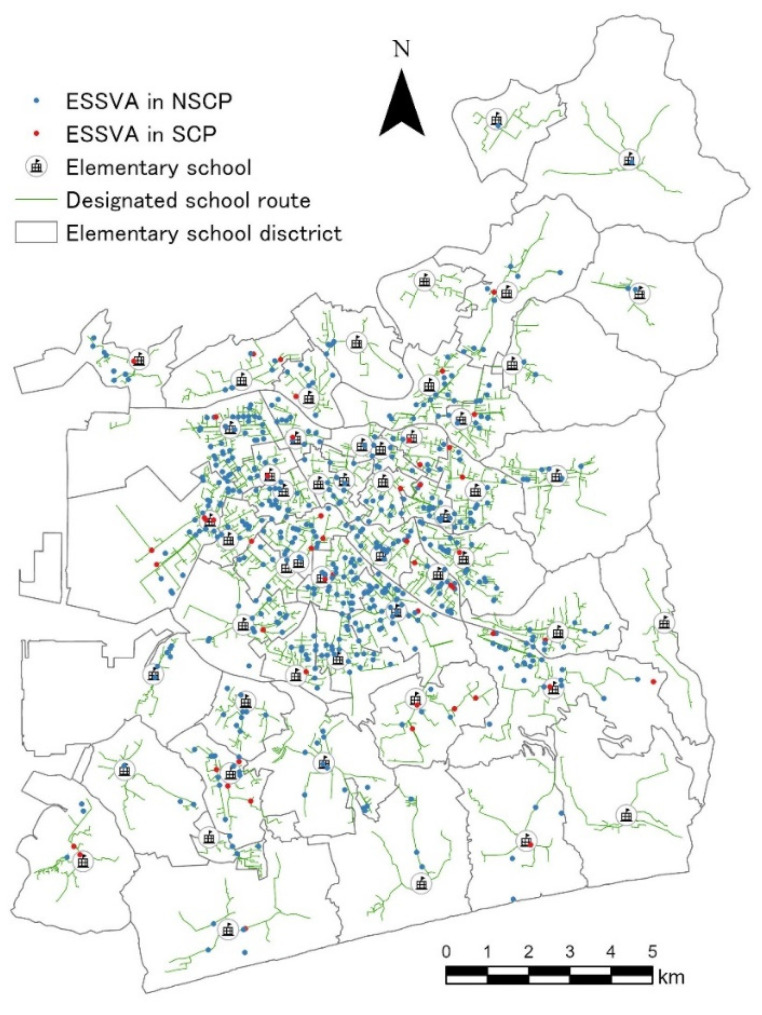
Locations of ESSVAs and designated school routes in the target area.

**Figure 2 ijerph-19-14123-f002:**
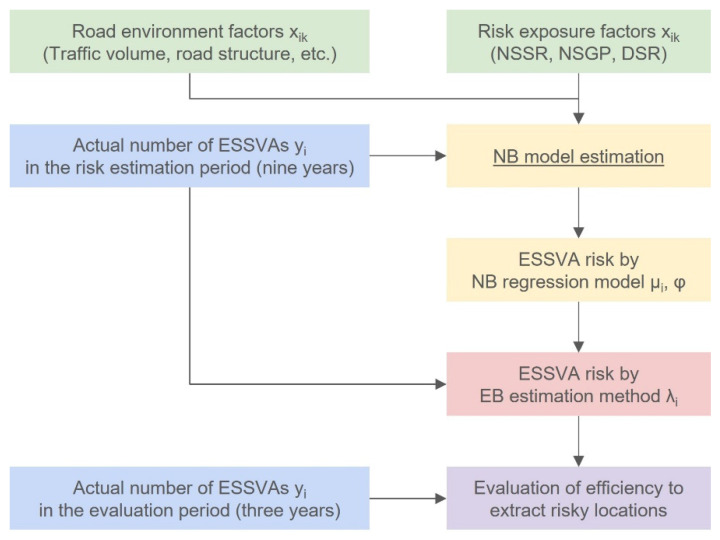
The flow of this study.

**Figure 3 ijerph-19-14123-f003:**
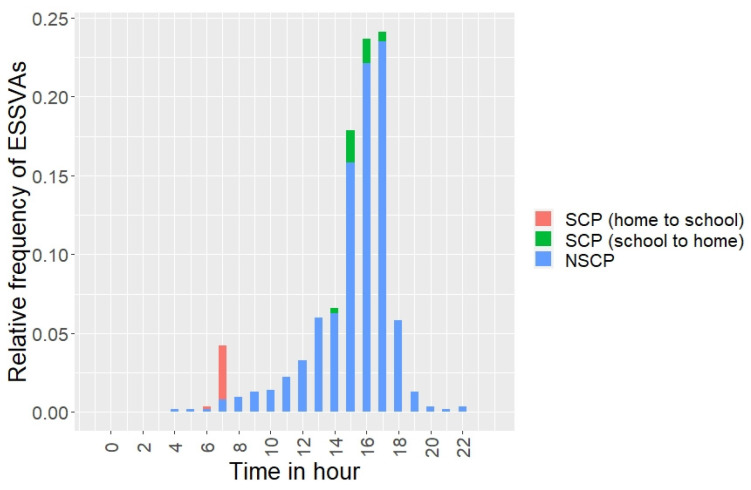
Time distribution of the occurrences of ESSVAs in the target area by travel purpose.

**Figure 4 ijerph-19-14123-f004:**
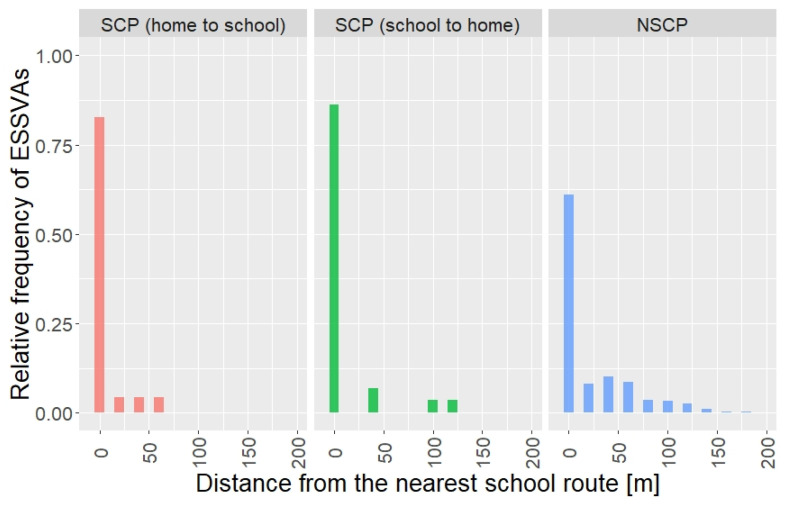
Distribution of the occurrences of ESSVAs in target area by the distance from the nearest school route and travel purpose.

**Figure 5 ijerph-19-14123-f005:**
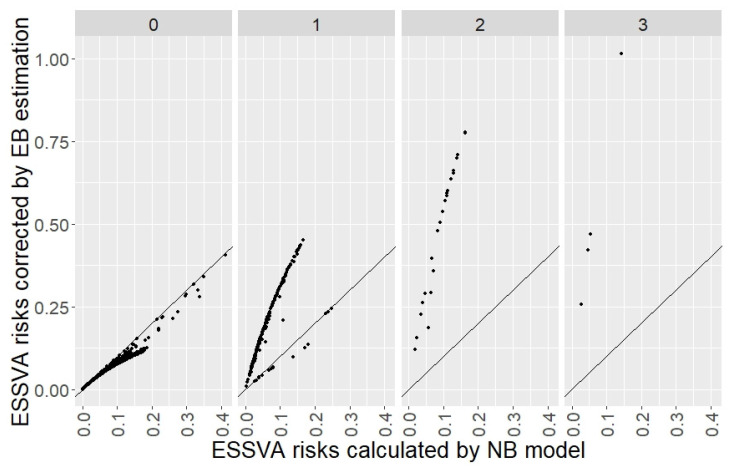
Comparison of ESSVA risks (the expected numbers of ESSVAs) according to the actual number of ESSVAs (0–3 from left to right) in the risk estimation period (nine years).

**Figure 6 ijerph-19-14123-f006:**
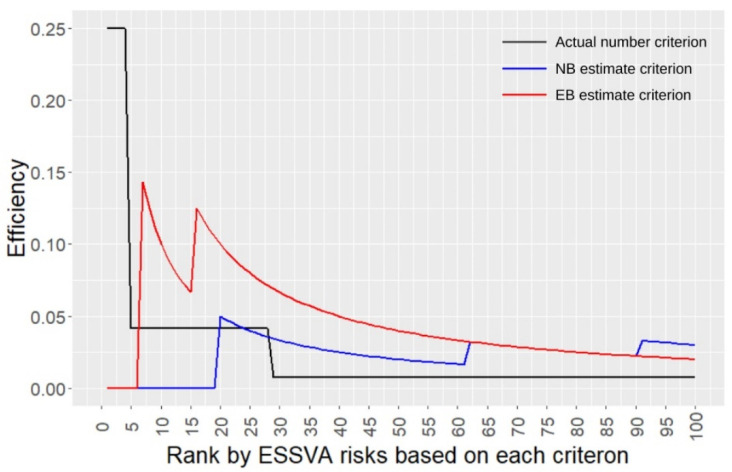
Comparison of the efficiencies for risky location extraction among the three criteria.

**Table 1 ijerph-19-14123-t001:** Main previous studies using a statistical modeling approach for the assessment of accident risk involving children by areas/locations.

Study	Unit of Analysis and Sample Size	Dependent Variable	Exposure	Other Factors	Method/Model	Country
[15]	Households 79 cases/ 110 controls	Presence/absence of crashes PMVCs ^1^ 5–17 age children 2008–2012	Number of trips/tours by questionnaire Walking time by questionnaire	Travel (daily activity) pattern Residential neighborhood type Age/Gender Parents’ work/Income	Case–control study Logistic regression model	Israel
[16]	Road segments 92 cases/ 368 controls Intersections 107 cases/ 428 controls	Presence/absence of crashes PMVCs ^1^ Weekdays, September to June, 7 a.m.–5 p.m. 5–14 age children 2002–2011	Child activity estimated by journey allocation model based on shortest route/ preferred route/ population	Intersection control type Crossing guard Average traffic flow Speed limit/One-way Land use Within 150 m of schools Road structures	Case–control study Logistic regression model	Canada
[17]	546 intersections	Number of perceived crash risk 10–12 age children 2015 Number of crashes PMVCs ^1^ 2007–2014	Children crossing estimated by questionnaire	Population density Street density Park Student facility Road structures Traffic calming Building Land use	Negative binomial regression model Zero-inflated negative binomial regression model	Korea
[18]	5703 road segments	Presence/absence of crashes PMVCs ^1^ Within 0.25 mile from schools 5–19 age children 2010–2014	Child population density	Bus stop Road class Road structures Land use Race	Logistic regression model	USA
[19]	School attendance boundaries 50 case/ 50 control	Highest/lowest quartile of crashes rate PMVCs ^1^ School travel time crashes 4–12 age children 2000–2013	Proportion of children walking to school by observation at schools	One-way Crossing guard Traffic light Traffic calming Land use Higher school disadvantage Inner suburbs/downtown	Case–control study Logistic regression model	Canada
[20]	47 prefectures	Killed or seriously injured children (KSI) rate Elementary Pedestrian 6–12 years Junior high bicycle and pedestrian 12–15 years	Child population	Travel purpose Proportion of population in DID ^3^	Multiple linear regression model	Japan
This study	7719 intersections	Number of crashes ESSVAs ^2^ 2007–2014”	Number of students on the surrounding school routes/gathering points Distance to the nearest school route	Travel purpose Road structures Traffic light Park Land use DID ^3^	Negative binomial regression model Empirical Bayes estimation	Japan

^1^ Pedestrian-motor vehicle crashes. ^2^ Elementary school students-vehicle accidents. ^3^ Densely inhabited district.

**Table 2 ijerph-19-14123-t002:** Estimation results of the NB regression models for the ESSVAs.

Explanatory Variable	SCP Model		NCSP Model	
	$\mathrm{Parameter}\;b_{k}$	$exp\;\left(b_{k}\right)$	$\mathrm{Parameter}\;b_{k}$	$exp\;\left(b_{k}\right)$
Constant term	−9.77 ***	0.00006	−4.78 ***	0.00840
Children’s risk exposure:				
NSSR15 ^1^	0.0129 ***	1.013		
DSR ^2^			−0.00771 ***	0.992
Natural logarithm of probe vehicle pass count	0.342 *	1.035	0.198 ***	1.22
Number of intersection legs (reference to three legs):				
four legs			0.717 ***	2.05
five or more legs			0.399	1.50
Distance to the nearest park ≤ 200 m (reference to more than 200 m)	−0.976 ^#^	0.377		
Area land use (reference to other land uses):				
high-rise buildings	3.87 **	47.9	0.461	1.59
low-density low-rise buildings	1.91 ^#^	6.75	0.594 **	1.81
high-density low-rise buildings	3.75 **	42.5	0.291	1.34
DID areas (reference to non−DID areas):	−1.32 ^#^	0.267		
Sample size (number of intersections)	7719		7719	
McFadden’s likelihood ratio	0.27		0.12	

^1^ number of elementary school students on school routes within 15 m of the intersection. ^2^ distance to the nearest school route from the intersection. ***: *p* < 0.001; **: *p* < 0.01; *: *p* < 0.05; ^#^: *p* < 0.1.

## Data Availability

Not applicable.

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
