# Peer review of "A Method for Locational Risk Estimation of Vehicle–Children Accidents Considering Children’s Travel Purposes"

_ijerph, 2022, doi:10.3390/ijerph192114123_

Round 1

Reviewer 1 Report

Notice some minor typos in the paper (criteron, ESSAV).

In the text (Line 320), rename Figure 7 to Figure 6.

Please explain Figures 5 and 6 more clearly.

In explaining statistical models, they are theoretical, without the insight on application. Can the authors connect the formulas with the explined models more precisely?

Author Response

We thank you for thoughtful questions and comments on our manuscript. We have revised our manuscript in accordance with your comments. The changes are listed in the below. We hope that our responses meet your expectations.

Point 1: Notice some minor typos in the paper (criteron, ESSAV).

Response 1: Thank you for your pointing typos. We have checked and revised typos throughoutly.

Point 2: In the text (Line 320), rename Figure 7 to Figure 6.

Response 2:  Thank you for your pointing it. We have revised it.

Point 3: Please explain Figures 5 and 6 more clearly.

Response 3:  We agree with the advice. We have added more explanation in Section 2.4 and Section 3.3 to be understandable regarding Figure 5 and Figure 6. We have also modified Figure 6 to have the legend.

Point 4: In explaining statistical models, they are theoretical, without the insight on application. Can the authors connect the formulas with the explined models more precisely?

Response 4:  We agree with the advice. We have added the explanation at the final paragraph in Section 2.3.3 to be connecting the explanation of model formulas with the application.

Reviewer 2 Report

1. The research problem of the paper is very meaningful, which is an important field in the field of traffic safety. 2. The current paper can be further improved in terms of data analysis and charts. For example, the existing correlation analysis is a bit simple, and the presentation form of some charts can be further enhanced. 3. Is the prediction algorithm of the paper universal? The traffic environment in different countries is quite different.

Author Response

We thank you for thoughtful questions and comments on our manuscript. We have revised our manuscript in accordance with your comments. The changes are listed in the below. We hope that our responses meet your expectations.

Point 1: The current paper can be further improved in terms of data analysis and charts. For example, the existing correlation analysis is a bit simple, and the presentation form of some charts can be further enhanced.

Response 1: We agree with the advice. We have added more explanation in Section 2.4 and Section 3.3 to be understandable regarding Figure 5 and Figure 6. We have modified Figure 6 to have the legend. We have also modified some texts in Figure 4.

Point 2: Is the prediction algorithm of the paper universal? The traffic environment in different countries is quite different.

Response 2: Thank you for your important question. As you pointed,  the road traffic environment conditions and its data availability differ among regions or countries. Therefore, although the same statistical modeling and predicting approach can be used for other regions and countries (i.e., NB regression model and EB estimation method), it is necessary to consider an objective variable and a variety of explanatory variables including risk exposure indicators by tailoring to those conditions and data availability. We have added a mention regarding this issue as the third limitations at the final paragraph in Section 4.

Reviewer 3 Report

A developed paper. It is well researched and put together. The methodology is sound and well demonstrated.

My suggestions:

 In table 1 the studies must be indicated with the number of the reference; e.g. [15] and not Elias and Shiftan (2014); Table 1 is not clear, please improve the format.

Acronyms/Abbreviations should be defined the first time they appear in the text; acronyms such as PMVC, DID used in the Table 1 are not defined.

Author Response

We thank you for thoughtful questions and comments on our manuscript. We have revised our manuscript in accordance with your comments. The changes are listed in the below. We hope that our responses meet your expectations.

Point 1: In table 1 the studies must be indicated with the number of the reference; e.g. [15] and not Elias and Shiftan (2014); Table 1 is not clear, please improve the format.

Response 1: We agree with the advice. We have replaced the sudies names with the numbers of references. We have also improved the format of Table 1 to make it easier to see by adding the dashed row lines.

Point 2: Acronyms/Abbreviations should be defined the first time they appear in the text; acronyms such as PMVC, DID used in the Table 1 are not defined.

Response 2: We agree with the advice. We have checked and revised the difinisions of abbreviations throughoutly. Also, regarding the abreviations in Table 1, we have added the difinisions by footnotes with footnote numbers.